# Gut Microbial Signatures Associated with Cryptosporidiosis: A Case Series

**DOI:** 10.3390/microorganisms13020342

**Published:** 2025-02-05

**Authors:** Antonia Piazzesi, Stefania Pane, Lorenza Romani, Francesca Toto, Matteo Scanu, Riccardo Marsiglia, Federica Del Chierico, Nicola Cotugno, Paolo Palma, Lorenza Putignani

**Affiliations:** 1Management and Diagnostic Innovations & Clinical Pathways Research Area, Unit of Microbiome, Bambino Gesù Children’s Hospital, IRCCS, 00146 Rome, Italy; afelicia.piazzesi@opbg.net (A.P.); francesca.toto@opbg.net (F.T.); matteo.scanu@opbg.net (M.S.); riccardo.marsiglia@opbg.net (R.M.); federica.delchierico@opbg.net (F.D.C.); 2Unit of Microbiology and Diagnostic Immunology, Unit of Microbiomics, Bambino Gesù Children’s Hospital, IRCCS, 00146 Rome, Italy; stefania.pane@opbg.net; 3Infectious Diseases Unit, Bambino Gesù Children’s Hospital, IRCCS, 00146 Rome, Italy; lorenza.romani@opbg.net; 4Unit of Clinical Immunology and Vaccinology, Bambino Gesù Children’s Hospital, IRCCS, 00165 Rome, Italy; nicola.cotugno@opbg.net (N.C.); paolo.palma@opbg.net (P.P.); 5Department of Systems Medicine, University of Rome “Tor Vergata”, 00133 Rome, Italy; 6Unit of Microbiology and Diagnostic Immunology, Unit of Microbiomics and Management and Diagnostic Innovations & Clinical Pathways Research Area, Unit of Microbiome, Bambino Gesù Children’s Hospital, IRCCS, 00146 Rome, Italy

**Keywords:** *Cryptosporidium*, cryptosporidiosis, chronic immunodeficiency, gut microbiota, metataxonomic sequencing

## Abstract

*Cryptosporidium* spp. are zoonotic protozoan parasites with a global prevalence, with both gastrointestinal and pulmonary involvement. Though symptoms can often be relatively mild, they can become severe and even fatal in children under five, the elderly, and in immunocompromised individuals, making cryptosporidiosis a leading cause of morbidity and mortality in fragile populations. Furthermore, there is an urgent clinical need for alternative therapies against cryptosporidiosis, as currently available FDA-approved treatments are ineffective in the immunocompromised. Recent evidence in animal models suggests that the gut microbiota (GM) can influence both host and parasite biology to influence the course of *Cryptosporidium* infection. Here, we present GM profiles in five cases of cryptosporidiosis, associated with varying underlying pathologies. We found that moderate–severe cryptosporidiosis was characterized by a reduction in alpha-diversity and an enrichment of *Enterococcus* spp., while decreases in *Bifidobacterium*, *Gemmiger*, and *Blautia* were detectable in the milder manifestations of the disease. Our results suggest that severe cryptosporidiosis is associated with a stronger change on the GM than is age or underlying pathology. Together with previously published studies in animal models, we believe that these results suggest that the GM could be a potential therapeutic target for human patients as well, particularly in the immunocompromised for whom anti-*Cryptosporidium* treatment remains largely ineffective.

## 1. Introduction

*Cryptosporidium*, Tyzzer, 1907 (Eucoccidiorida: Cryptosporidiidae), is a genus comprising over 40 species of protozoan parasites, of which over 20 can also infect humans [1,2]. Infection begins with the ingestion of oocysts, most commonly from contaminated water, food, or from direct animal–human or human–human contact. Once they reach the small intestine, sporozoites are released and attach themselves to the host enterocytes, forming an intracellular but extra-cytoplasmic vacuole in which the sporozoite differentiates into a trophozoite [3]. Mitotic division of the trophozoite produces merozoites, which can behave like sporozoites by escaping the vacuole, attaching to neighboring enterocytes, and beginning the process over again. Alternatively, merozoites can also undergo sexually dimorphic differentiation and reproduce sexually, producing diploid zygotes which differentiate into oocysts. These oocysts can then either continue to re-infect the host, or be released into the environment to infect other individuals [3,4,5].

Once infected, humans most commonly present with watery diarrhea, vomiting, fever, abdominal pain, and coughing [3,4,5,6]. Though many are able to cope with infection within a two-week period, even without medication, very young children, the elderly and the immunocompromised are at an increased risk of severe morbidity and mortality. One study found that, in 2016, cryptosporidiosis was globally responsible for over 48,000 deaths and a loss of 12.05 million disability-adjusted life-years in children under five, making it the fifth leading diarrheal etiology in the world [6]. Another study conducted in Africa and Asia reported even more dire numbers, finding approximately 202,000 *Cryptosporidium*-attributable deaths in children under two, a net increase of 59,000 deaths compared to *Cryptosporidium*-negative children with similar symptoms [7].

Immunocompromised individuals are at the highest risk of morbidity and mortality due to chronic cryptosporidiosis, largely due to the ineffectiveness of anti-*Cryptosporidium* treatment in patients with low CD4+ cell counts [3,8]. To this day, cryptosporidiosis remains a highly dangerous infection for the immunocompromised, including primary immunodeficiencies, people with cancer, and those with auto-immune disorders or transplant recipients who require pharmacological immunosuppressive therapy [3,9,10].

The commensal gut microbiota (GM) has long been recognized as a highly dynamic microbial community which, when targeted, can precipitate drastic change to human health. Modulations of the GM, either by dietary intervention, probiotic supplementation, or fecal microbiota transplantation (FMT), have yielded promising results in the context of gastrointestinal diseases [11,12,13,14,15], cancer [16,17,18], metabolic disorders [19,20,21,22], and infectious diseases [23,24,25]. While GM modulations in the context of parasitic infections have been less studied, preliminary evidence suggests that there is significant interplay between the GM, the immune system, and human parasites, making GM manipulation a promising alternative to currently available antiparasitic therapies [26,27,28,29,30,31,32].

In the context of cryptosporidiosis, preclinical studies and case reports indicate that GM modulation could represent a promising therapeutic alternative to currently available anti-*Cryptosporidium* drugs. In immunosuppressed mice, probiotic treatment helped mice clear parasitic infection, indicating that, unlike the currently available therapies on the market, GM modulation does not require a functioning immune system to combat cryptosporidiosis. Instead, GM modulation is hypothesized to act by indole metabolism on the parasite mitosome [33,34,35,36]. In humans, probiotic treatment was found to be helpful in combatting symptoms of *Cryptosporidium* infection in immunocompetent children [37,38]. However, to our knowledge, no studies in immunocompromised human patients have been conducted to investigate the potential of GM modulations in combatting chronic cryptosporidiosis. Moreover, the interplay between the GM and *Cryptosporidium* is still not fully described or understood.

Previously, we published the first case report of the effects of chronic cryptosporidiosis on the GM of an immunocompromised child [39]. However, due to this being a single case, we were not able to determine whether the differences we observed were due to the child’s primary immunodeficiency, his liver failure, or chronic *Cryptosporidium* infection. Here, we profiled the GM of five patients infected with *Cryptosporidium*, four of whom were immunodeficient, and one of whom was immunocompetent, in order to see whether common GM signatures associated with cryptosporidiosis emerged. *Cryptosporidium*-induced GM modulations could indicate the most promising bacterial targets for alternative or adjuvant cryptosporidiosis therapies, particularly for immunocompromised patients with few clinical options in combatting the disease.

## 2. Materials and Methods

### 2.1. Patient Selection

The patients presented in this case series were admitted to Bambino Gesù Children’s Hospital in Rome, Italy, for issues pertaining to their own clinical history. Patients who were positive for *Cryptosporidium* infection and who provided samples for GM characterization were included in this case series. All included patients or their legal guardians provided informed consent for anonymized publication of their clinical data. Furthermore, given that all patients included in this study were of different ages, and given that age has an effect on GM composition [40], GM profiles of each patient were compared to a group of 5 age-matched healthy subjects (controls; CTRLs). CTRLs were enrolled during an epidemiological survey carried out at the Microbiome Unit of Bambino Gesù Children’s Hospital (BBMRI Human Microbiome Biobank, OPBG) to generate a reference biobank of samples from healthy subjects. Healthy CTRLs were defined as volunteers with no known pathologies or food allergies, within the healthy BMI range, who have not had any infectious disease, antibiotics, or probiotics in the month prior to sample collection, and who agreed to donate their stool samples for biobanking and anonymized use in clinical studies. The use of healthy subjects in this study was approved by the Ethical Committee of the Bambino Gesù Children’s Hospital, IRCCS (protocol No. 1113_OPBG_2016) and was conducted in accordance with the Principles of Good Clinical Practice and the Declaration of Helsinki.

### 2.2. Cryptosporidium Detection in Fecal Smears by Optical Microscopy

Fecal smears were prepared for *Cryptosporidium* detection as previously described [41] with slight modifications. Fecal samples were collected, filtered with a 10% formalin disposable system (Parapak, 10% formalin fixative, Meridian Biosciences, Cincinnati, OH, USA) and then prepared for microscopic observation with a modified Ziehl–Neelsen staining protocol (Becton Dickinson, Franklin Lakes, NJ, USA). A thin fecal smear was deposited onto slides, air dried, then fixed in methanol for 5 min. The slides were incubated with Carbol fuchsin stain (Sigma-Aldrich, Darmstadt, Germany) for 30 min, rinsed with tap water and incubated with acid alcohol (1%) for 30 s. Then, slides were rinsed with tap water, incubated with methylene blue(Sigma-Aldrich, Darmstadt, Gerany) for 1 min, rinsed again with tap water and left to air dry. Each slide was examined (using oil immersion) under 100× magnification using a light microscope [42]. At least 10 fields for each of at least 5 slides were examined to perform morphometry and to count *Cryptosporidium* spp. oocysts in order to produce an average number of oocysts for each slide. Overall, approximately ten slides were imaged to produce a quantitative estimation of parasite oocysts for each time-point of the quantitative titration confirmed by a second operator.

### 2.3. Cryptosporidium Detection by PCR

DNA was extracted from stool samples with the automated QIASymphony system as per the manufacturer’s instructions (Qiagen, Hilden, Germany). The presence of *Cryptosporidium* spp. DNA was confirmed with the AllPlexTM Gastrointestinal Parasite Assay as per the manufacturer’s instructions (Seegene, Seoul, Republic of Korea).

### 2.4. Clinical Evaluation of Cryptosporidiosis Severity

Once *Cryptosporidium* infection was confirmed by microscopy and molecular biology methods, patients were categorized as either mild, moderate, or severe cases based on a clinical evaluation of patient symptoms based on World Health Organization guidelines [43]. Patients with severe clinical manifestations were defined as those with chronic diarrhea, i.e., patients passing three or more loose stools per day for over 30 days. Patients with mild presentation were those with diarrhea for less than 14 days, and moderately affected patients were defined as those passing three or more loose stools per day for over 14 days but for less than one month.

### 2.5. Stool Sample Collection

Stool samples were collected with sterile stool collection kits and stored at 4 °C until transfer to the Microbiome Unit of the Bambino Gesù Children’s Hospital. Samples were then aliquoted and stored at −80 °C until DNA extraction.

### 2.6. 16S rRNA Targeted Metataxonomics

16S rRNA-based sequencing was performed as previously described [44]. Briefly, DNA was extracted from stool samples with a QIAmp Fast DNA Stool mini kit (Qiagen, Hilden, Germany) and the V3-V4 hypervariable region was amplified with a 2× KAPA Hifi HotStart ready Mix (KAPA Biosystems Inc., Wilmington, MA, USA) according to the manufacturer’s instructions. Illumina Nextera adaptor primers were used to index the samples, after which they were sequenced on an Illumina MiSeq Platform (Illumina, San Diego, CA, USA).

### 2.7. Bioinformatics Analysis

Paired-end fastq files were pre-processed as patient–age-matched CTRL batches using the Quantitative Insights Into Microbial Ecology 2 (QIIME2) v2024.5 bioinformatics pipeline [45]. The q2—DADA2 plugin was used to perform quality checks (QC) of reads and to produce the Amplicon Sequence Variants (ASVs) [46]. Each ASV was classified taxonomically by means of the q2—Greengenes plugin using the nucleotide database Greengenes2 v2022.10 [47]. Finally, phylogenetic analysis was performed with the q2—phylogeny align-to-tree-mafft-fasttree plugin obtaining rooted trees [45].

### 2.8. Statistical Analyses

Count tables, taxonomy tables, and phylogenetic trees of each sample were imported in R v4.4.1 for statistical analyses. In ecological analyses, all count tables were normalized with the rarefaction method based on the minimum sample depth. For alpha-diversity analysis, Shannon–Weiner, Simpson, and Chao-1 indices were calculated and Mann–Whitney U-tests were used to compare bacterial diversity between the two independent groups. Count tables were merged obtaining a unique count table in order to compare distance dissimilarity, calculated with the Bray–Curtis algorithm, between all groups. Hierarchical cluster analysis was performed with the Pvclust R package (version 2.2) based on the bootstrap approach (replications: 1000). To evaluate differential bacterial abundances, each count table was normalized with the Cumulative Sum Scaling (CSS) method [48] and transformed into relative abundances.

## 3. Results

### 3.1. Case Description

*Case 1:* The first patient, hereafter referred to as Crypto 1, was a 9-year-old child diagnosed with HIGM due to a mutation in the gene encoding for CD40L. Crypto 1 was also suffering from chronic diarrhea for three years due to a severe and chronic Cryptosporidium parvum infection that persisted even after multiple rounds of antiparasitic treatment, which precipitated sclerosing cholangitis and hepatic cirrhosis. Due to his severe liver damage and his primary immunodeficiency, Crypto 1 was admitted to undergo a liver transplant followed by a bone marrow transplant. Seven days after the liver transplant, while the patient was undergoing antibiotics and yet another round of antiparasitic treatment, we collected four stool samples over a one-week period, and selected five samples from age-matched controls (CTRLs) for comparison (Table 1).

*Case 2:* Crypto 2 was 26 years old, also affected by HIGM due to a genetic defect in CD40L, who had been admitted to our hospital following complaints of diarrhea and abdominal pain beginning 30 days prior, and had been undergoing antibiotic treatment as a result (Table 1). After a mild case of cryptosporidiosis was diagnosed by microscopic and molecular means, we collected three consecutive stool samples from this patient and compared them with five age-matched CTRLs.

*Case 3:* Crypto 3 was a 14-year-old child affected by systemic lupus erythematosus (SLE), and was undergoing aggressive immunosuppressive therapy as a result. During this treatment, Crypto 3 acquired a gastrointestinal XDR Acinetobacter infection, for which he was undergoing antibiotic treatment and was being screened for a possible experimental FMT, in order to combat this infection and avoid sepsis, a common complication and cause of mortality in patients with SLE. During screening, Crypto 3 was also diagnosed with moderate–severe Cryptosporidium infection, upon which time the stool samples collected for screening underwent 16S rRNA metataxonomic sequencing and compared with five age-matched CTRLs.

*Case 4:* Crypto 4 was a 12-year-old child affected by sterile alpha motif domain containing 9 like (SAM9DL) primary immunodeficiency (SAMD9L–Ataxia–Pancytopenia Syndrome, SAPS). He was admitted to our hospital with gastrointestinal symptoms, including vomiting and diarrhea, for 10 days prior to admission. Screening for infections revealed he was positive for severe Cryptosporidium infection, and thus was not given antibiotics at the time of stool sample collection for 16S targeted metataxonomic sequencing.

*Case 5:* Crypto 5 was an immunocompetent, 6-year-old child with no underlying conditions, admitted because of abdominal pain, after which he was diagnosed with a mild case of cryptosporidiosis. Before treatment commenced, three stool samples were collected for 16S targeted metataxonomic sequencing and compared to five healthy age-matched CTRLs.

### 3.2. Patients with Moderate or Severe Cryptosporidiosis Present with Reduced Alpha-Diversity and Increased Enterococcus spp. Compared with CTRLs

We profiled the GM of our five patients by performing next-generation sequencing of the V3-V4 hypervariable region of the bacterial 16S rRNA gene on two to four stool samples collected from each patient. We then compared the metataxonomic profiles of each patient to a group of five age-matched CTRLs.

First, we measured GM alpha-diversity by calculating Simpson’s index for each patient and CTRL sample (Figure 1). We found that our three moderately–severely affected patients, namely Crypto 1 (Figure 1A), Crypto 3 (Figure 1C), and Crypto 4 (Figure 1D), all had reduced species richness in their GM when compared to their own age-matched CTRLs. On the other hand, our two mildly affected patients Crypto 2 (Figure 1B) and Crypto 5 (Figure 1E) did not have reduced alpha-diversity compared to CTRLs, suggesting that cryptosporidiosis is associated with reduced GM species richness only in the more severe manifestations of the disease.

We next compared patient GM metataxonomic profiles to CTRL subjects, starting with those patients who presented with moderate–severe cryptosporidiosis. We found that, in all samples profiled from moderately–severely affected patients, one single bacterial genus predominated in the GM, with relative abundances of close to or over 90% (Figure 2). We found that Crypto 1 samples were predominantly composed of *Enterococcus* spp., composing between 84 and 98% of the total GM of this patient (Figure 2A), followed by *Bacteroides H* and *Streptococcus*. In two Crypto 3 samples, we found an overwhelming predominance of *Klebsiella* (Figure 2B) which, after sample collection, would also be diagnosed systemically in this patient. However, similar to Crypto 1, we also found a large relative abundance of *Enterococcus* spp. in one out of the three replicates, which was almost completely absent in CTRLs (Figure 2B). Interestingly, Crypto 4 were also found to be predominantly composed of *Enterococcus* spp., consistent with the results observed in Crypto 1 and Crypto 3 (Figure 2C), and despite having yet another kind of underlying immunodeficiency, and despite not having been treated with antibiotics, as the other two moderate–severely affected patients were (Table 1). The GM of age-matched CTRLs, on the other hand, were far more diverse, and were primarily composed of *Bifidobacterium* spp., *Gemmiger* spp., *Akkermansia* spp., and *Blautia* spp. (Figure 2A). These results largely explain the reduced alpha-diversity measurements illustrated in Figure 1. Taken together, these results seem to suggest that an overabundance of *Enterococcus* spp. is a common GM signature in patients with moderate to severe *Cryptosporidium* infection, regardless of underlying pathology, or antibiotic treatment.

### 3.3. Mild Cases of Cryptosporidiosis Were Associated with a More Diverse GM

We next profiled the GM of our two mildly affected patients, each compared to their own group of age-matched CTRLs (Figure 3). We found that, unlike severely affected patients, those with a mild case of cryptosporidiosis had a far more diverse GM, consistent with the alpha-diversity measurements presented in Figure 1. We found a similar bacterial community structure between Crypto 2 and CTRLs, being mostly composed of *Phocaeicola* spp., *Akkermansia* spp., *Prevotella* spp., *Gemmiger* spp., and *Bacteroides* spp. (Figure 3A). Similarly, the taxonomic composition of the GM of Crypto 5 was far more diverse and far more similar to CTRLs than that observed in their severely affected counterparts (Figure 3B). The GM of Crypto 5 was composed primarily of *Phocaeicola* spp. and *Bacteroides* spp., as well as *Faecalibacterium* spp. and *Bifidobacterium* spp. (Figure 3B. The most notable differences between these two groups, however, was a decrease in *Gemmiger* spp. and an increase in *Agathobacter* spp. and *Sutterella* spp. in Crypto 5 compared with CTRLs (Figure 3B). Taken together, these results suggest that the more severe forms of cryptosporidiosis are associated with larger GM changes than are their mildly affected counterparts.

### 3.4. Patients with Severe Cryptosporidiosis Cluster Separately from Those with Mild Infection

We next decided to see whether the GM profiles of patients infected with *Cryptosporidium* would cluster by severity, by antibiotic treatment, or with their own age-matched CTRLs. However, in order to obtain accurate GM maps on a patient-by-patient basis, our analyses thus far have involved collecting multiple samples per patient, in order to ensure that the GM signatures we observe were consistent. This also results in less variability between samples collected from the same patient, compared to the variability one would observe between samples collected from different subjects. Therefore, in order to ensure that these differences in variability are not producing statistical artefacts, we performed a Bray–Curtis analysis only considering the first sample collected from each patient (Figure 4).

Bray–Curtis dissimilarity produced a very separate and distinct cluster composed of the moderately–severely affected patients Crypto 1, 3 and 4 (Figure 4). On the other hand, mildly affected patients, namely Crypto 2 and Crypto 5, clustered more closely with CTRLs than with those with severe cryptosporidiosis (Figure 4). These results were particularly interesting in light of the fact that one severely affected patient (Crypto 4) was not undergoing antibiotic treatment at the time of sample collection, while one mildly-affected patient (Crypto 2) was. Despite this, Crypto 4 samples still clustered tightly with the antibiotic-treated and severely affected Crypto 1. Taken together, these results are in line with our previous observations, and indicate that patients with severe cryptosporidiosis more closely resemble each other than they do healthy subjects, despite the differences in their ages, underlying pathologies, and history of antimicrobial treatment. Furthermore, these results show that patients with mild cryptosporidiosis are more similar to CTRLs than patients with a heavier parasitic burden, suggesting that *Cryptosporidium* infection might affect GM ecology in a dose-dependent manner.

### 3.5. Common Gut Microbial Signatures of Cryptosporidium Infection

As observed above, we found that all three patients with moderate–severe cryptosporidiosis had a GM that was heavily dominated by *Enterococcus* spp. (Figure 2). Given the fact that these three patients all had very different underlying pathologies and ongoing treatment plans, these results seem to suggest that an overabundance of *Enterococcus* spp. is a common signature of severe cryptosporidiosis infection (Figure 2). However, our two patients with mild cryptosporidiosis did not have even a trend of increased *Enterococcus* abundance in their GM (Figure 3), which may suggest that this finding presents itself in those affected by moderate to severe infection.

Along with a drastic increase in *Enterococcus*, our severely affected patients also had drastically reduced relative abundances of *Akkermansia*, *Bifidobacterium*, *Gemmiger*, and *Blautia* (Figure 2). However, it is important to note that, when the GM of an individual is predominantly colonized by any single bacterial genus, mathematically, this will lead to all other abundant genera being reduced. Therefore, it is difficult to discern which of these reductions may be biologically relevant, and which are simply a statistical artefact given the high relative abundance of *Enterococcus*. Indeed, while *Akkermansia* is reduced in Crypto 1, 3, and 4, it is actually increased in Crypto 2, and was highly variable among the three replicates of Crypto 5 (Figure 3). Given these results, we cannot discern whether the reduction in *Akkermansia* observed in Crypto 1, 3, and 4 is a statistical artefact, or whether it is a biologically relevant event tied to *Enterococcus* expansion. Having said this, we did observe a substantial decrease in *Bifidobacterium*, *Gemmiger*, and *Blautia* in Crypto 5 as well as in Crypto 1, 3, and 4, despite the fact that the GM of Crypto 5 was not dominated by a single bacterial genus (Figure 2 and Figure 3). These results might suggest that, while an overabundance of *Enterococcus* may be a signature of moderate to severe cryptosporidiosis, reductions in *Bifidobacterium*, *Gemmiger*, and *Blautia* may be early indicators of *Cryptosporidium* infection.

Finally, we found that *Agathobacter* was substantially enriched in Crypto 5, though no trend towards this effect was observed in any other patient (Figure 2 and Figure 3). Further studies are needed to confirm whether this bacterial genus may be a marker for mild *Cryptosporidium* infection in immunocompetent patients, or whether it was a peculiarity of this specific subject.

## 4. Discussion

Last year, we presented a case report of a child with chronic cryptosporidiosis and CD40L immunodeficiency, whose GM was characterized by an increase in *Enterococcus*, *Prevotella*, and *Campylobacter* [39]. Although this was the first report of GM modulations in response to severe *Cryptosporidium* infection, it was extremely preliminary, insofar as it was impossible to determine which GM signatures may have been due to the child’s parasitic infection, his primary immunodeficiency, or his liver complications. Here, we present five patients with different underlying pathologies and ages, to see whether there are commonalities between the GMs of this otherwise diverse group of patients.

The most striking commonality found between the three patients with the more severe forms of cryptosporidiosis was a predominance of *Enterococcus* spp. It is important to note that the *Enterococcus* genus is comprised of many multi-drug-resistant species, and as such can increase substantially in the GM of patients given multiple rounds of antibiotics [49]. Though antibiotics will not act on *Cryptosporidium* directly, they are often used to treat the gastrointestinal symptoms commonly associated with cryptosporidiosis. Having said this, only two of the three moderately–severely affected patients had been treated with antibiotics prior to stool sample collection, as well as one of our mildly affected patients (Table 1). Indeed, as shown in Figure 4, patients who had been treated with antibiotics did not cluster together. Crypto 2, a patient treated with antibiotics and with a mild case of cryptosporidiosis, was placed by our analyses in the center of a cluster of healthy CTRLs. Furthermore Crypto 4, a moderately–affected patient with no antibiotic use prior to sample collection, clustered closely with his two severely-affected and antibiotics-treated counterparts (Figure 4). Therefore, while the increase of *Enterococcus* spp. in response to antibiotics has been described, antibiotic use in our patient set did not overlay with those with a predominance of *Enterococcus*. While these therapies may have contributed to the increase in *Enterococcus* observed in some patients, it does not fully explain the patterns that we observe. Furthermore, these results seem to suggest that cryptosporidiosis exerts a stronger influence on the GM than does antibiotics treatment, though further studies on larger patient cohorts are needed to confirm this.

In our patient dataset, all patients with moderate–severe cryptosporidiosis were characterized by an overabundance of *Enterococcus* spp. in the GM, while the two patients with mild clinical cases were not. However, despite the fact that they had very different underlying pathologies, these three patients also shared two other important clinical characteristics. The first of these is a severe immunodeficiency which, though caused by different clinical circumstances, does raise the possibility that immunodeficiency itself is enough to cause an overabundance of *Enterococcus* spp. in the gut. On the one hand, it is very difficult to separate the two, because immunodeficiency is one of the main factors which precipitates severe and chronic cryptosporidiosis and other infections, causing them to often be treated with, among other things, multiple rounds of antibiotics. On the other hand, there have been publications on the gut microbial composition of people with AIDS [50,51], and common variable immunodeficiency [52,53,54,55], none of which report a common signature of *Enterococcus* overabundance. While most of these studies have been conducted in adults, and therefore insight into the effects of immunodeficiency on pediatric GM composition are still somewhat lacking, the current literature does not suggest that immunodeficiency commonly produces a GM signature rich in *Enterococcus* spp.

The second common clinical characteristic shared by these three patients is multiple episodes of diarrhea, as this is one of the clinical criteria for distinguishing between mild and severe cases of cryptosporidiosis [43]. Once again, the scientific literature does not suggest a predominance of *Enterococcus* spp. as a common, universal signature of multiple diarrheal episodes. For example, one study of patients with diarrheal-predominant Irritable Bowel Syndrome (IBS-D) found a decrease of Firmicutes, the phylum in which *Enterococcus* spp. are classified, and an expanded population of Proteobacteria in IBS-D patients compared to healthy CTRLs [56]. Other studies on post-cholecystectomy diarrhea (PCD) found a slight but significant increase in *Enterococcus* spp. despite an overall decrease in the relative abundance of Firmicutes, accompanied by a large increase in Bacteroidota [57,58]. In pediatric cohorts with infectious diarrhea, studies described an overabundance of Proteobacteria, as well as a predominance of *Streptococcus* spp. in the GM [59,60,61,62]. However, given the wide range of ages and geographical regions sampled in the aforementioned papers, future studies on larger cohorts of *Cryptosporidium*-positive patients may also include diarrheal samples of different etiology, in order to control for changes in GM composition due to frequent, watery bowel movements.

Though some bacterial genera, such as *Bifidobacterium*, are largely associated with positive health outcomes, *Enterococcus* is a more diverse genus in the GM. On the one hand, it is one of the first genera to populate the human GM, and is responsible for the production of various vitamins and metabolites necessary for human health [63]. On the other hand, this genus comprises species that can become highly virulent and infective, and have been associated with increased inflammation, systemic infection, and meningitis [63,64,65,66]. However, in animal studies, probiotic administration of *Enterococcus faecalis* CECT 7121 has been found to interfere with parasitosis, including in immunosuppressed mouse models of cryptosporidiosis [33,67]. Similarly, *Enterococcus faecium* CCM8558 and *Enterococcus durans* ED26E/7 were found to have antiparasitic activity against *Trichinella spiralis* infection in mice, further underscoring the beneficial potential of some *Enterococcus* strains in combatting parasitic infection [68]. Unfortunately, the methodology used in this study was not able to identify which species of *Enterococcus* were present in these patients, which could be elucidated with future studies.

Another interesting commonality identified between these patients was a marked reduction in the genera *Bifidobacterium*, *Gemmiger* and *Blautia* (Figure 2 and Figure 3). While this is to be expected in patients Crypto 1, 3 and 4, given that their GMs were over 80% populated by either *Enterococcus* or *Klebsiella*, this reduction was also observed in Crypto 5, who did not have a strongly altered GM compared to age-matched CTRLs (Figure 3B). In Crypto 2, there was more variability between samples, making it impossible to conclude whether or not there was a trend in reduction of these three genera in this patient (Figure 3A). On one hand, it is important to note that Crypto 2 was the only adult patient profiled in this study, and *Bifidobacterium* is known to possess a less important role in the GMs of adults than in that of children [69,70]. On the other hand, given that this is only one patient, we cannot, at this time, discern whether a reduction of *Bifidobacterium* is a signature of pediatric cryptosporidiosis, or whether Crypto 2 possessed other characteristics that explain these findings. Future studies are necessary to determine whether or not these differences between Crypto 2 and pediatric patients are due to different GM alterations in response to adult *Cryptosporidium* infection, or whether it is due to this individual’s own GM variability.

*Bifidobacterium*, as a whole, is widely recognized as a genus of health-promoting bacteria, which is why it is a very popular genus for the development of probiotic strains, including those designed to combat parasitic infections [71,72,73]. Indeed, one in vitro study found that the supernatants from both *Bifidobacterium* and *Lactobacillus* strains could inhibit *Cryptosporidium* oocyst viability [73]. In this context, it is possible that *Cryptosporidium* either more readily infects patients with low *Bifidobacterium* abundances, or that it capable of reducing the relative abundances of those bacteria which are more detrimental to its survival. While future longitudinal studies into GM markers for *Cryptosporidium* predisposition could be very informative, our results seem to suggest that, in children, low levels of *Bifidobacterium* in the GM are measurable in even in the milder stages of cryptosporidiosis.

The genera *Blautia* and *Gemmiger* are far less studied than *Bifidobacterium*, especially in the context of parasitic infection. Generally speaking, *Blautia* is positively correlated with metabolic health [74,75], while *Gemmiger* has been found to be negatively associated with protozoan-induced diarrhea in livestock [76]. Though further studies into the roles of these two genera in parasitic infection would be highly informative, our results still seem to suggest that, as observed with *Bifidobacterium*, reduced relative abundances in these two genera may be common GM signatures of *Cryptosporidium* infection in children, even in the mild stages of the disease.

The last variable to consider when interpreting these results is the potential influence of different *Cryptosporidium* species and genotypes on the GM of immunocompromised patients. So far, over 20 *Cryptosporidium* species capable of infecting humans have been described, though the two most common species to infect humans are *C. parvum* and *C. hominis*. Furthermore, each *Cryptosporidium* species is further subdivided into genotypes and subgenotypes [77]. However, most clinical protocols for the diagnosis of *Cryptosporidium* infection do not resolve past the genus level and, therefore, there is incomplete data on the prevalence of these genotypes outside of epidemiological studies, and even less knowledge about the potential infectivity of different *Cryptosporidium* subtypes. Indeed, in this study, only Crypto 1 was known to be infected with *C. parvum*, while we do not possess this information for the other 4 patients in this Case Series. We cannot therefore exclude the possibility that different *Cryptosporidium* species or genotypes modulate the GM in different ways, and/or produce more or less severe manifestations of cryptosporidiosis in infected patients. Though such analyses would be beyond the scope of this study, particularly in light of our small sample size and case-by-case description, we do believe that genotyping *Cryptosporidium* during studies on larger cohorts may shed light on whether some subtypes, and therefore some parasitic reservoirs, pose a higher risk to vulnerable patients than others.

To our knowledge, this study is only the second description of GM modulations in response to cryptosporidiosis in immunocompromised patients. Given the commonalities found between these otherwise clinically diverse patients, we believe that the field would greatly benefit from further studies on larger patient cohorts, as this sample size is too small to be able to draw firm correlations between disease severity and GM composition. Should common GM signatures of cryptosporidiosis be confirmed and found, the medical community should consider investigating the potential of GM modulations in combatting cryptosporidiosis, particularly in light of what has been reported in preclinical mouse models.

To date, the only FDA-approved anti-*Cryptosporidium* drugs available rely on the host’s ability to produce CD4+ cells, which is why chronic cryptosporidiosis remains a real threat to immunocompromised patients [3,8]. However, studies in immunodeficient mouse models has shown that GM modulations can bypass the host’s immune system in combatting parasitosis [33,34]. Therefore, if consistent GM responses to mild or severe disease are found in immunodeficient patients with *Cryptosporidium* infections, these may lead researchers towards which subsequent GM modulations may be most beneficial to patients. Probiotic development, or even FMT, may have potential in aiding to combat this parasitic infection in vulnerable patients.

## 5. Conclusions

This is the first study to investigate how the immunocompromised GM responds to cryptosporidiosis in more than one patient. Given the differences between these patients in terms of age, underlying pathology, co-morbidities and antibiotics treatment, the fact that their GM profiles are so similar is indeed striking. Though this area of research could benefit greatly from a wider range of patients, these results suggest that severe *Cryptosporidium* infection has a greater effect on GM composition than does age, immune status, antibiotics use, or other organ involvement, both in terms of species richness and *Enterococcus* overabundance. This, together with the aforementioned preclinical studies, is further evidence that modulations of the GM may be a promising alternative to those antiparasitic drugs that fail to aid immunocompromised patients. Further studies are needed on larger cohorts of immunocompromised patients affected by cryptosporidiosis, not only to confirm these findings, but also to aid in identifying those microbial signatures which could help to develop the alternative therapies needed to protect these vulnerable populations.

## Figures and Tables

**Figure 1 microorganisms-13-00342-f001:**
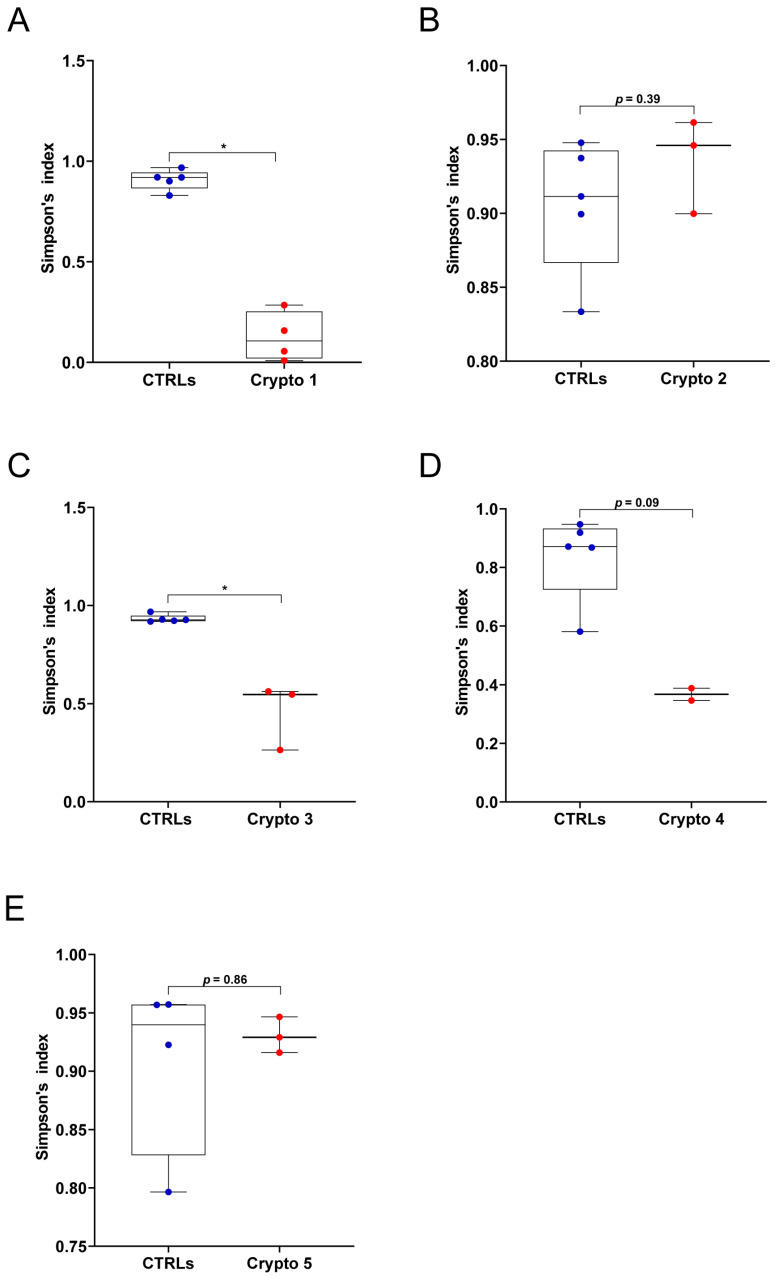
GM alpha-diversity in patients with mild and moderate–severe Cryptosporidium infections. (**A**–**E**) Alpha-diversity, as measured by Simpson’s index, in Crypto 1 (**A**), Crypto 2 (**B**), Crypto 3 (**C**), Crypto 4 (**D**) and Crypto 5 (**E**), compared to age-matched CTRLs. Statistical analysis: Mann–Whitney U-test, * *p* < 0.05. Patients with moderate to severe cryptosporidiosis (**A**,**C**,**D**) were characterized by a reduction in alpha-diversity measurements, while mildly affected patients (**B**,**E**) had comparable alpha-diversity measurements compared to age-matched CTRLs. In red: individual *Cryptosporidium*-positive replicates. In blue: individual CTRL replicates.

**Figure 2 microorganisms-13-00342-f002:**
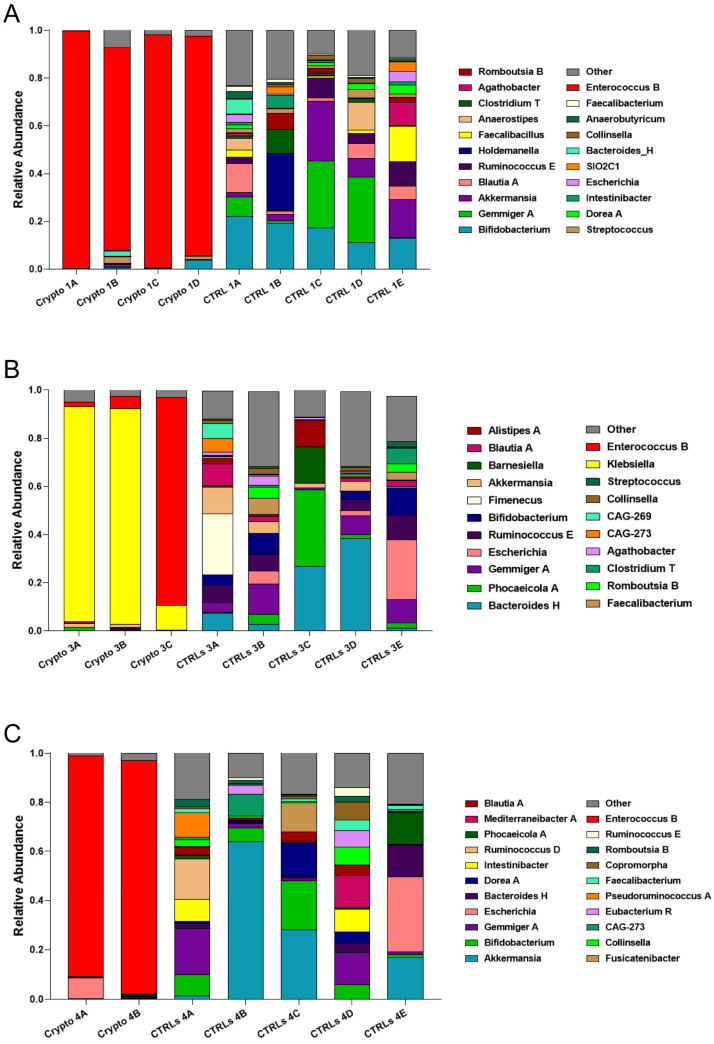
GM composition of patients with moderate–severe cryptosporidiosis. (**A**–**C**) Stacked bar charts representing the relative abundance of bacterial genera in each Crypto 1 (**A**), Crypto 3 (**B**), and Crypto 4 (**C**) replicate and each individual age-matched CTRL. Genera detected under 1% relative abundance were grouped together and represented collectively as “other”. Moderately–severely affected patients were characterized by an overabundance of a single bacterial genus in all profiled samples.

**Figure 3 microorganisms-13-00342-f003:**
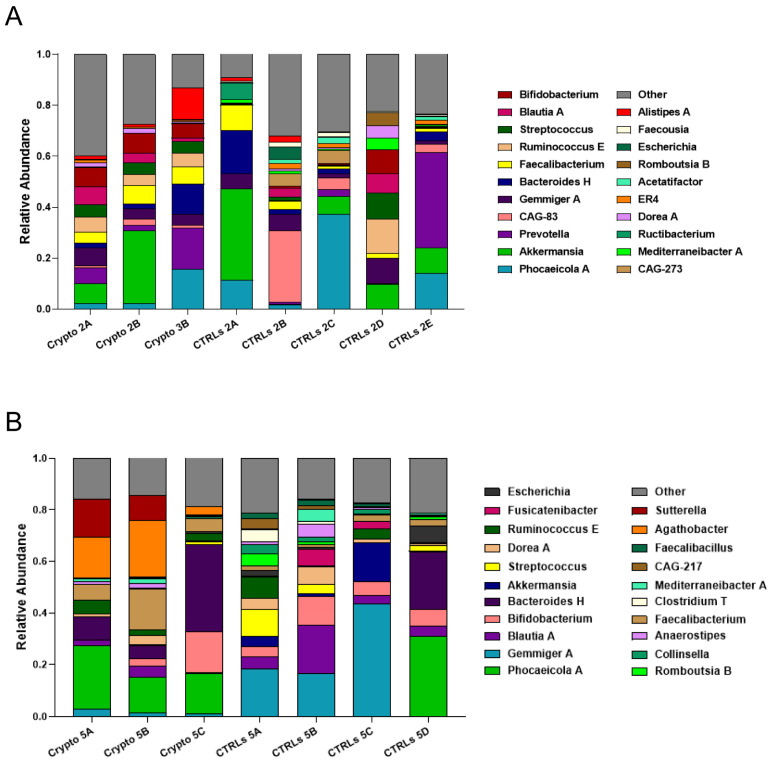
GM composition of patients with mild cryptosporidiosis. The GM of mildly affected patients are diverse and somewhat similar to that of their respective healthy CTRLs. (**A**,**B**) Stacked bar charts representing the relative abundance of bacterial genera in each Crypto 2 (**A**) and Crypto 5 (**B**) replicate and each individual age-matched CTRL. Genera detected under 1% relative abundance were grouped together and represented collectively as “other”.

**Figure 4 microorganisms-13-00342-f004:**
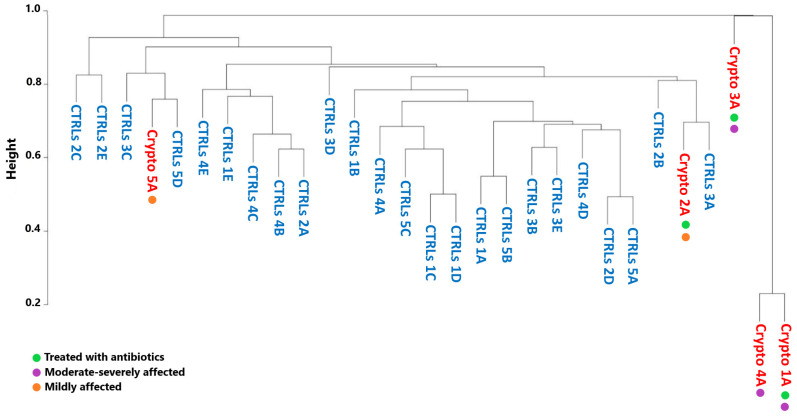
Beta-diversity analyses of Crypto-positive and CTRL samples. Dendrogram representing similarity between healthy CTRLs (blue) and a single sample from each *Cryptosporidium*-positive patient (red) as calculated by Bray–Curtis dissimilarity. Green: patients treated with antibiotics. Purple: patients with moderate–severe cryptosporidiosis. Orange: patients with mild cryptosporidiosis.

**Table 1 microorganisms-13-00342-t001:** Clinical characteristics of patients.

Subject	Age	Sex	Underlying Pathology	Cryptosporidiosis Severity	Coinfections	Antibiotic Treatment	Other Treatment
Crypto 1	8	M	CD40L primary immunodeficiency	Severe	None	Meropenem, Amikacin	Liposomal amphotericin B, Azithromycin, Nitazoxanide, Tacrolimus
Crypto 2	26	M	CD40L primary immunodeficiency	Mild	None	Trimethoprim	Fluconazole
Crypto 3	14	M	Systemic lupus erythematosus	Severe	XDR *Acinetobacter*	Meropenem, Tigecycline	None
Crypto 4	12	M	SAMD9L-Ataxia-Pancytopenia Syndrome	Moderate	None	None	None
Crypto 5	6	M	None	Mild	None	None	None

## Data Availability

The datasets analyzed in this study have been deposited in the NCBI Sequence Read Archive (SRA) under the accession number PRJNA1144251 and will be set to private until the acceptance of this manuscript. A reviewer’s link to the data will be provided upon request.

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
