# Peer review of "Gut Microbial Signatures Associated with Cryptosporidiosis: A Case Series"

_microorganisms, 2025, doi:10.3390/microorganisms13020342_

Round 1

Reviewer 1 Report (Previous Reviewer 2)

Comments and Suggestions for Authors

The authors followed the reviewer's recommendation and adjusted the manuscript.

However, the  ,, Materials and methods'' begins directly with the presentation of the fecal collection. Please add some sentences about the study design and about the patiens from this study. You presented the patiens at the Results without presenting few words about them also in Materials and methods. 

Author Response

Comment 1: The authors followed the reviewer's recommendation and adjusted the manuscript.

Response 1: We would like to thank the reviewer for their appreciation of our revision.

Comment 2: However, the  ,, Materials and methods'' begins directly with the presentation of the fecal collection. Please add some sentences about the study design and about the patiens from this study. You presented the patiens at the Results without presenting few words about them also in Materials and methods. 

Response 2: We would like to thank the reviewer for their suggestion and have updated the Materials and Methods accordingly. We have also moved the paragraph about healthy controls from the Case Description to this section of the Materials and Methods.

Reviewer 2 Report (New Reviewer)

Comments and Suggestions for Authors

Abstract:

    • The role of the gut microbiota (GM) in cryptosporidiosis needs to be stated more clearly to contextualize the results for a broader audience.
    • A concluding sentence summarizing the potential impact of the findings would strengthen the abstract.

Introduction:

    • The discussion of gut microbiota (GM) and its modulation is insightful but could be streamlined to focus more on the study's direct relevance to cryptosporidiosis.
    • More detail about the gaps in knowledge this study aims to address would help justify the research.

Results:

    • Figures (e.g., Figure 1) effectively illustrate differences in GM composition but could benefit from more detailed captions explaining their significance.
    • The clustering analysis is intriguing but doesn't discuss potential confounding factors like antibiotic use in enough detail.
    • The relationship between clinical severity and GM changes is suggested but not explored in depth.

Discussion:

    • The overabundance of Enterococcus spp. is highlighted as a common signature of severe cryptosporidiosis, but potential biases (e.g., small sample size, antibiotic treatment) are not adequately addressed.
    • The discussion about Bifidobacterium and its reduction is compelling but needs stronger evidence linking this to cryptosporidiosis rather than other factors.
    • There is limited discussion on how these findings could translate into clinical applications or therapies.

Conclusion:

    • It reiterates results rather than providing novel insights or actionable recommendations.
    • There is no mention of limitations or suggestions for overcoming them in future research.

Author Response

Comment 1: Abstract:

The role of the gut microbiota (GM) in cryptosporidiosis needs to be stated more clearly to contextualize the results for a broader audience.

Response 1: We have added a sentence to the abstract to highlight the connection between the GM and parasitic infection as per the Reviewer’s request.

Comment 2: A concluding sentence summarizing the potential impact of the findings would strengthen the abstract.

Response 2: We have modified the concluding sentence of the Abstract in order to strengthen the message we would like to transmit to the readers.

Comment 3: Introduction:

The discussion of gut microbiota (GM) and its modulation is insightful but could be streamlined to focus more on the study's direct relevance to cryptosporidiosis.

Response 3: We have removed some sentences from the Introduction that are not directly relevant to Cryptosporidium in order to streamline it.

Comment 4: More detail about the gaps in knowledge this study aims to address would help justify the research.

Response 4: We have expanded this part of the Introduction in order to comply with the Reviewer’s request.

Comment 5: Results:

Figures (e.g., Figure 1) effectively illustrate differences in GM composition but could benefit from more detailed captions explaining their significance.

Response 5: We have added a summarizing sentence to each of the Figure captions to summarize the significance of the results. We have also expanded the Results sections pertaining to those figures in order to further highlight the significance of the findings.

Comment 6: The clustering analysis is intriguing but doesn't discuss potential confounding factors like antibiotic use in enough detail.

Response 6: We have added additional information on the patients who were treated with antibiotics and those who were not in the Results, including the Case Description and in the paragraphs discussing the clustering analysis, in order to further clarify the significance of the clustering analysis.

Comment 7: The relationship between clinical severity and GM changes is suggested but not explored in depth.

Response 7: Given that this is a Case Series, and therefore, by definition, a study on a very small number of patients, we believe that the relationship between clinical severity and GM changes cannot be explored more deeply with this particular dataset. However, the fact that there were so many similarities between such otherwise clinically diverse patients was, in our opinion, the most surprising and suggestive result uncovered in this study, which led to our belief that these results should be published, particularly in light of the dearth of published information on the subject.

Comment 8: Discussion:

The overabundance of Enterococcus spp. is highlighted as a common signature of severe cryptosporidiosis, but potential biases (e.g., small sample size, antibiotic treatment) are not adequately addressed.

Response 8: We have discussed and elaborated on the potential influence of other factors on our results, such as sample size (L511-514; L515-519), antibiotic treatment (L398-415), diarrhea (L432-447), immunodeficiency (L420-431) and Cryptosporidium species (L497-514), in the Discussion.

Comment 9: The discussion about Bifidobacterium and its reduction is compelling but needs stronger evidence linking this to cryptosporidiosis rather than other factors.

Response 9: We agree with the Reviewer that these results are compelling but will need to be confirmed on larger patient cohorts to be conclusive, and we have reiterated this in the Discussion and Conclusions as well (L414-415, L474-477, L511-514, L517-519, L535-540).

Comment 10:  There is limited discussion on how these findings could translate into clinical applications or therapies.

Response 10: We have expanded the final paragraph of the Discussion in order to discuss these considerations in more depth, as per the Reviewer’s suggestion.

Comment 11: Conclusion:

It reiterates results rather than providing novel insights or actionable recommendations.

Response 11: In the Conclusions, we argue that our results indicate that the medical field would benefit from further studies into the GM’s response to cryptosporidiosis on larger cohorts, in an effort to pinpoint which bacteria would provide the best targets for pharmacological GM modulations, especially since there are no FDA-approved anti-Cryptosporidium treatment options available for the immunocompromised. We have also added a sentence to the beginning of the Conclusions underlining the novelty of the manuscript itself.

Comment 12: There is no mention of limitations or suggestions for overcoming them in future research.

Response 12: We have discussed the limitations, and how to overcome those limitations, regarding sample size (L511-514; L515-519), antibiotic treatment (L398-415), diarrhea (L432-447), immunodeficiency (L420-431) and Cryptosporidium species (L497-514), in the Discussion.

Reviewer 3 Report (New Reviewer)

Comments and Suggestions for Authors

The paper intends to relate infection with Cryptosporidium to a decrease in bacterial diversity of the gut.

1)        The paper does not indicate if the five patients were infected with the same cryptosporidium species or not. Over 23 Cryptosporidium species/genotypes have been identified in humans, and C. hominis and C. parvum are the most common species (more than 90%) responsible for human cryptosporidiosis. The differences observed could be due to different Cryptosporidium species/genotypes.

2)        The choice of patients is somewhat critically important: they all have (except no 5) other underlying pathologies, most are treated with antibiotics (except no 4 and 5), one has a coinfection with Acinetobacter (no 3). There is thus quite a big diversity within these five patients and besides they are also suffering from mild to severe Cryptosporidiosis. It is the very difficult to assess the role of cryptosporidiosis on the bacterial diversity. It would be also interesting to know the status of the controls: if they are patients from hospital they probably have also some pathology affecting them.

3)        The use of the Bray-Curtis similarity index is common for evaluating beta-diversity. It is however sensitive to undersampling. It generally results in poor accuracy and leads to underestimates of assemblage similarities. See: Hardersen, S., & La Porta, G. (2023). Never underestimate biodiversity: how undersampling affects Bray–Curtis similarity estimates and a possible countermeasure. The European Zoological Journal, 90(2), 660–672. https://doi.org/10.1080/24750263.2023.2249007. With three samplings of the same Cryptosporidium patient vs five non-infected persons, we are clearly in the undersampling. The comparisons are then probably underestimating alpha-diversity.

4)        It is alpha diversity then not beta diversity if you compare the three samples from one patient. It is a mixture of alpha and beta diversity if you integrate the five non-infected with the three samplings of  infected patients.

5)        Mann-Whitney non-parametric test is intended for independent samples and with at least three individuals. How could you compare patient 1 (three related samples) with his age adjusted five controls (independent samples)?

6)        Finally, the data are not driving to a compelling conclusion.  This small and heterogeneous sample of patients is overanalysed. I wonder how the dendrograms will remain similar when submitted to jackknife or bootstrap resampling.

Author Response

Comment 1: The paper intends to relate infection with Cryptosporidium to a decrease in bacterial diversity of the gut.

1)        The paper does not indicate if the five patients were infected with the same cryptosporidium species or not. Over 23 Cryptosporidium species/genotypes have been identified in humans, and C. hominis and C. parvum are the most common species (more than 90%) responsible for human cryptosporidiosis. The differences observed could be due to different Cryptosporidium species/genotypes.

Response 1: Crypto 1 was infected with C. parvum, which we have now included in the Case Description section. However, unfortunately, we did not perform analyses on the other 4 patients that would permit us to determine the Cryptosporidium species involved. We have added a paragraph to the Discussion to address the possibility that different Cryptosporidium species may elicit different GM responses.

Comment 2: 2)        The choice of patients is somewhat critically important: they all have (except no 5) other underlying pathologies, most are treated with antibiotics (except no 4 and 5), one has a coinfection with Acinetobacter (no 3). There is thus quite a big diversity within these five patients and besides they are also suffering from mild to severe Cryptosporidiosis. It is the very difficult to assess the role of cryptosporidiosis on the bacterial diversity.

Response 2: We agree with the Reviewer that the patient group is very heterogeneous, which is why we decided to present these results as a Case Series and analyze the results on a case-by-case basis, rather than group them together and risk having this clinical diversity mar the results. Indeed, it is precisely their heterogeneity which leads us to believe that these results are highly suggestive and of potential interest to the Microorganisms readership, as they appear to suggest that Cryptosporidium infection is associated with GM profiles that transcend age, antibiotic treatment, or underlying pathology. Having said this, we have stated clearly that these results, being the first of their kind, are preliminary, suggestive, and will need to be repeated on much larger patient cohorts before they can be taken as conclusive of Cryptosporidium-induced GM modulations. However, this study, in our opinion, has produced results that are suggestive enough to justify the investment needed for future follow-up studies, which is why we believe that they should be shared with the scientific community.

Comment 3: It would be also interesting to know the status of the controls: if they are patients from hospital they probably have also some pathology affecting them.

Response 3: The healthy CTRLs used in this study were collected from healthy volunteers, with no known pathologies and no antibiotic or probiotic use in the month leading up to sample collection, and were collected as part of a separate epidemiological study. This description has now been moved to the first paragraph of the Materials and Methods section. Furthermore, additional information has been added to this paragraph in order to further clarify how healthy CTRLs were selected.

Comment 4: 3)        The use of the Bray-Curtis similarity index is common for evaluating beta-diversity. It is however sensitive to undersampling. It generally results in poor accuracy and leads to underestimates of assemblage similarities. See: Hardersen, S., & La Porta, G. (2023). Never underestimate biodiversity: how undersampling affects Bray–Curtis similarity estimates and a possible countermeasure. The European Zoological Journal, 90(2), 660–672. https://doi.org/10.1080/24750263.2023.2249007. With three samplings of the same Cryptosporidium patient vs five non-infected persons, we are clearly in the undersampling. The comparisons are then probably underestimating alpha-diversity.

Response 4: In this manuscript, we used Simpson’s index to measure alpha diversity, and Bray-Curtis dissimilarity to measure beta-diversity and produce dendrograms indicating the degree of similarity between samples. We also repeated this analysis by using only one sample per Cryptosporidium-positive patient, in order to ensure that our results were not a statistical artefact of repeated sampling within the same patient.

Furthermore, we have also performed this analysis with the UnWeighted UniFrac algorithm, but we initially removed this from the manuscript when we needed to reduce the number of Figures in order to fit the Brief Report format. This analysis also produced very similar results to the Bray-Curtis dissimilarity analysis included in the paper, which is why we felt that they were redundant and removed them from this version of the manuscript. However, in light of the Reviewer’s request, we have re-inserted these results into the manuscript as a Supplementary Figure instead.

Comment 5: 4)        It is alpha diversity then not beta diversity if you compare the three samples from one patient. It is a mixture of alpha and beta diversity if you integrate the five non-infected with the three samplings of infected patients.

Response 5: Alpha-diversity measurements, such as the Simpson’s index used in this study, are used to estimate species richness, i.e. the number of bacterial taxa, found in each group. Simpson’s index is calculated for each individual sample, based on a rarefaction curve to control for the number of sequencing reads present in each sample. After these calculations were performed, we compared the median of the Simpson’s indicies calculated for the patient samples with the median of the Simpson’s indicies calculated for their respective CTRL group, with a Mann Whitney-U test. We did not integrate these two groups together, nor did we perform comparisons within these groups. The algorithms used for calculating alpha and beta diversity are completely different from one another.

Comment 6: 5)        Mann-Whitney non-parametric test is intended for independent samples and with at least three individuals. How could you compare patient 1 (three related samples) with his age adjusted five controls (independent samples)?

Response 6: We agree with the Reviewer that the Mann-Whitney U test is used for independant samples, and is not for paired datasets, such as before and after measurements. However, in this case, there is no relationship between the two groups that are being compared and where the statistical test is being applied, i.e. between the Crypto and the CTRL samples. While we can concede that samples collected from the same individual may tend to be less variable than samples collected between multiple individuals, we chose to use a Mann-Whitney U test to compare our two groups because 1) the use of non-parametric statistical analyses is the standard in the field, 2) non-parametric tests make fewer assumptions, and 3) we are unaware of a more appropriate statistical analysis we could have used in this context. Furthermore, non-parametric statistical analyses are characterized by being less statistically powerful than parametric ones, and are thus more likely to produce false negative rather than false positive results. Indeed, if we apply a parametric test to these data, we obtain much smaller p-values. Given all of these considerations, we feel that this test is the only one that we may reasonably apply to our data.

Comment 7: 6)        Finally, the data are not driving to a compelling conclusion.  This small and heterogeneous sample of patients is overanalysed. I wonder how the dendrograms will remain similar when submitted to jackknife or bootstrap resampling.

Response 7: As stated in the Statistical Analysis section of the Materials and Methods, bootstrapping was applied to this analysis. Furthermore, as mentioned above, we have included dendrograms created with UnWeighted UniFrac analysis as a Supplementary Figure, which are also consistent with the results of our Bray-Curtis analysis. Given the consistency in the results after the use of these different methods, as well as the clear overabundance of Enterococcus spp observed in our 3 severely affected patients, we believe that the data do indeed suggest that the GM of patients with severe cryptosporidiosis resemble each other more than they do healthy CTRLs of the same age group, or antibiotic-treated mildly affected patients.

Reviewer 4 Report (New Reviewer)

Comments and Suggestions for Authors

The manuscript deals with the morphological and PCR detection of Cryptosporidium infection in humans in Rome (Italy) and the impact of this parasitic infection on the gut microbiota community. The article is interesting, is devoted to relevant theme and contain new information. It contains statistical analysis and is well illustrated. The statistical component is the strong point of the work. The manuscript of Italian colleagues fits into scope of Microorganisms and could be published after minor corrections.

The only thing that is confusing is that the sample of Cryptosporidium infected patients is too small, which does not allow us to draw deep conclusions. Moreover, all patients are different in their characteristics. The authors examined some specific cases of cryptosporidiosis, which again do not allow the material to be generalized to all people. But this is at the discretion of the Academic Editor.

Lines 275–287 – This information should be in the Materials and Methods. So how many people participated in the study overall? or how many people's data did you use?

And some small remarks:

1. It is necessary to return to the text (in the Materials and Methods section) information about where the study was conducted and where did the patients come from.

2. At the first mention of species or genus in text, its Latin name with the author and year of description should be given (in lines 35, 75, 330, 331, 722, 723, etc.).

And for the genus Cryptosporidium it is desirable to also provide the names of higher taxa – Cryptosporidium Tyzzer, 1907 (Eucoccidiorida: Cryptosporidiidae)

In conclusion, I express my opinion. The manuscript can be published after minor correction.

Author Response

Comment 1: The manuscript deals with the morphological and PCR detection of Cryptosporidium infection in humans in Rome (Italy) and the impact of this parasitic infection on the gut microbiota community. The article is interesting, is devoted to relevant theme and contain new information. It contains statistical analysis and is well illustrated. The statistical component is the strong point of the work. The manuscript of Italian colleagues fits into scope of Microorganisms and could be published after minor corrections.

Response 1: We would like to thank the Reviewer for their kind comment.

Comment 2: The only thing that is confusing is that the sample of Cryptosporidium infected patients is too small, which does not allow us to draw deep conclusions. Moreover, all patients are different in their characteristics. The authors examined some specific cases of cryptosporidiosis, which again do not allow the material to be generalized to all people. But this is at the discretion of the Academic Editor.

Response 2: We agree with the Reviewer that the patient group is very heterogeneous, which is why we decided to present these results as a Case Series and analyze the results on a case-by-case basis, rather than group them together and risk having this clinical diversity mar the results. Indeed, it is precisely their heterogeneity which leads us to believe that these results are highly suggestive and of potential interest to the Microorganisms readership, as they appear to suggest that Cryptosporidium infection is associated with GM profiles that transcend age, antibiotic treatment, or underlying pathology. Having said this, we have stated clearly that these results, being the first of their kind, are preliminary, suggestive, and will need to be repeated on much larger patient cohorts before they can be taken as conclusive of Cryptosporidium-induced GM modulations. However, this study, in our opinion, has produced results suggestive enough to justify the investment needed for future follow-up studies, which is why we believe that they should be shared with the scientific community.

Comment 3: Lines 275–287 – This information should be in the Materials and Methods. So how many people participated in the study overall? or how many people's data did you use?

Response 3: We have added a Patient Selection section to the beginning of the Materials and Methods section to clarify these points.

Comment 4: And some small remarks:

  1. It is necessary to return to the text (in the Materials and Methods section) information about where the study was conducted and where did the patients come from.

Response 4: We have added this information to the aforementioned Patient Selection section of the Materials and Methods.

 Comment 5: 2. At the first mention of species or genus in text, its Latin name with the author and year of description should be given (in lines 35, 75, 330, 331, 722, 723, etc.). And for the genus Cryptosporidium it is desirable to also provide the names of higher taxa – Cryptosporidium Tyzzer, 1907 (Eucoccidiorida: Cryptosporidiidae)

Response 5: We have added the requested nomenclature for Cryptosporidium. However, this is not standard practice for bacterial nomenclature, as described in Oren et. al. 2023 (https://doi.org/10.1099/ijsem.0.005585). We feel that adding this information for each bacterial genus will break up the text too much and make it less easy to read, as well as render the manuscript’s nomenclature inconsistent with the current literature.

Comment 6: In conclusion, I express my opinion. The manuscript can be published after minor correction.

Response 6: We would like to thank the Reviewer for their comments and for taking the time to review our manuscript

Round 2

Reviewer 2 Report (New Reviewer)

Comments and Suggestions for Authors

The points I pointed out have been well corrected. Thanks for your hard work.

Author Response

Comment 1: The points I pointed out have been well corrected. Thanks for your hard work.

Response 1: We would like to thank the Reviewer for their kind comment and for taking the time to review our manuscript

Reviewer 3 Report (New Reviewer)

Comments and Suggestions for Authors

The paper was improved by identifying better the controls. I  still have some problems with the paper.

1) The  ressults are over-analysed compared to the size of the group of patients. The number of figures is too large. Fig 1 ok. Fig 2: ok.   Fig 3.ok But keep only relative abundance and Shannon diversity index whatever the infection clinical status. Fig 4: keep B only.  For the next figure keep Bray-Curtis  with C only.  I do not think that supplementary figures are needed.

2)  The figure of signatures of Cryptosporidium infection could be deleted, since there is a large variability among controls as well. It could be only mentioned from the proportions of species. A signature is difficult to find since the major finding is a decrease of the number of species with a huge dominance of Enterococcus and Klebsiella.

3) Concerning alpha and beta diversity used by ecologists: alpha concerns diversity  in a site at local scale, beta diversity (differences between sites at a local scale) and gamma (diversity in several site at a lanscap size= a combination of alpha + beta diversity).  It is a bit tricky to use in the medical world. A site could be a patient either infected , then or not infected= one site each) and beta could be the different patients (infected or non infected: two sites). There can be  severali nterpretation. Explain why you use alpha and beta diversity.

3) The use of nonparametric statistics is a good choice for these data. If you use Mann and Whitney, you should mention that you consider that the patients are independant samples although the Crypto ones are not fully in dependant.

The main finding is that Cryptosporidium infection is linked with  a loss of diversity, that the patients are treated with antibiotics or not. The paper will benefit to concentrate on that finding.

Author Response

Comment 1: The paper was improved by identifying better the controls. I  still have some problems with the paper.

  • The  ressults are over-analysed compared to the size of the group of patients. The number of figures is too large. Fig 1 ok. Fig 2: ok.   Fig 3.ok But keep only relative abundance and Shannon diversity index whatever the infection clinical status. Fig 4: keep B only.  For the next figure keep Bray-Curtis  with C only.  I do not think that supplementary figures are needed.

Response 1: The Supplementary Figure was only included in order to respond to the Reviewer’s concern about bootstrapping and whether the results presented by the dendrogram were reliable. If the Reviewer is satisfied with the UnWeighted UniFrac results and doesn’t think the should include it in the final version, we are more than happy to remove these results from the paper again.

With regards to Figure 4, it is unclear from the comment whether the Reviewer is asking us to keep only panel B or panel C. However, given other Reviewer’s comments regarding dendrograms, we have decided to only include panel C, in the hopes that this Reviewer also agrees that this is an appropriate solution. We have also modified the text within the Results in order to conform with this change.

Comment 2: The figure of signatures of Cryptosporidium infection could be deleted, since there is a large variability among controls as well. It could be only mentioned from the proportions of species. A signature is difficult to find since the major finding is a decrease of the number of species with a huge dominance of Enterococcus and Klebsiella.

Response 2: we have done as the Reviewer has suggested, and removed Figure 5, only referring to Figures 2 and 3 in the text. 

Comment 3: Concerning alpha and beta diversity used by ecologists: alpha concerns diversity  in a site at local scale, beta diversity (differences between sites at a local scale) and gamma (diversity in several site at a lanscap size= a combination of alpha + beta diversity).  It is a bit tricky to use in the medical world. A site could be a patient either infected , then or not infected= one site each) and beta could be the different patients (infected or non infected: two sites). There can be  severali nterpretation. Explain why you use alpha and beta diversity.

Response 3: The use of alpha- and beta-diversity measurements are standard practice in clinical microbiome studies, as well as the ecological studies that the Reviewer describes. Indeed, the original definitions of alpha, beta and gamma diversity, introduced by Whittaker in 1960, that the Reviewer cites, have been somewhat revisited since the introduction of human microbiome studies. For example, as stated in Finotello et. al. (doi: https://doi.org/10.1093/bib/bbw119): “in the study of the human microbiota, alpha diversity is used to describe the compositional complexity of a single sample, whereas beta diversity is used to describe taxonomical differences between samples. To simplify diversity indices down to an intuitive, qualitative definition, we can state that a sample has high alpha diversity when it contains a high number of equally abundant species, and low diversity otherwise. When comparing two samples, beta diversity is high if they share few species and low if most of their species are in common”.

In this paper, we used alpha diversity to describe, as mentioned above, the compositional complexity of a sample collected from an infected patient, compared to the complexity of a sample collected from a healthy one. Other examples in the published literature that use these measures in this fashion can be found in Chen et. al., 2021, Gut (doi: 10.1136/gutjnl-2021-324090); Madison et. al., 2023, J Gerontol A Biol Sci Med Sci (doi: 10.1093/gerona/glad276); Putignani et. al., 2021, Eur J Gastroenterol Hepatol (doi: 10.1097/MEG.0000000000002050); Li et. al., 2017, Microbiome (doi: 10.1186/s40168-016-0222-x); Wan et. al., 2022, Gut (doi: 10.1136/gutjnl-2020-324015); and Vandeputte et. al., 2015, Gut (doi: 10.1136/gutjnl-2015-309618), among others.

Comment 4: The use of nonparametric statistics is a good choice for these data. If you use Mann and Whitney, you should mention that you consider that the patients are independant samples although the Crypto ones are not fully in dependant.

Response 4: While the samples within the Crypto patients are not independant of each other, they are independant from the CTRL samples, and this comparison is where the test is being applied. Having said this, we have added the requested phrase to the Materials and Methods in the Statistical Analysis section.

Comment 5: The main finding is that Cryptosporidium infection is linked with a loss of diversity, that the patients are treated with antibiotics or not. The paper will benefit to concentrate on that finding

Response 5: We would like to thank the Reviewer for their kind comment. We have highlighted this finding further in the Conclusions, in order to draw more attention to this finding.

This manuscript is a resubmission of an earlier submission. The following is a list of the peer review reports and author responses from that submission.

Round 1

Reviewer 1 Report

Comments and Suggestions for Authors

see attached

Reviewer 2 Report

Comments and Suggestions for Authors

You presented that Crypto 2 was 27 years old. Is there a mistake?

Why presented the age of the children only for the first 3 patients? For the paper uniformity you have to present all the datas for all the patients.

Please present the method for feces collection - the moment of the day, if the samples were stored before analysis, in which conditions, etc....

The findings from this manuscript are important, but the number of patients are quite low. So, please add a sentence at the end of the paper that future researches are nedeed to sustaine these findings.

Please add some conclusions to the paper, related to your findings. It is important to highlight the main ideas of the study.